# Preparation of Fe_3_O_4_-Ag Nanocomposites with Silver Petals for SERS Application

**DOI:** 10.3390/nano11051288

**Published:** 2021-05-13

**Authors:** Thi Thuy Nguyen, Fayna Mammeri, Souad Ammar, Thi Bich Ngoc Nguyen, Trong Nghia Nguyen, Thi Ha Lien Nghiem, Nguyen Thi Thuy, Thi Anh Ho

**Affiliations:** 1Vietnam Academy of Science and Technology, Graduate University of Science and Technology, 18 Hoang Quoc Viet, Cau Giay, Hanoi 10000, Vietnam; 2Institute of Physics, Vietnam Academy of Science and Technology, 18 Hoang Quoc Viet, Cau Giay, Hanoi 10000, Vietnam; ntbngoc@iop.vast.ac.vn (T.B.N.N.); trongnghia@iop.vast.ac.vn (T.N.N.); halien@iop.vast.ac.vn (T.H.L.N.); 3ITODYS, Université de Paris, CNRS, UMR 7086, 15 rue J-A de Baïf, 75013 Paris, France; Fayna.mammeri@univ-paris-diderot.fr (F.M.); ammarmer@univ-paris-diderot.fr (S.A.); 4Institute of Materials Science, Vietnam Academy of Science and Technology, 18 Hoang Quoc Viet, Cau Giay, Hanoi 10000, Vietnam; ntthuy@ims.vast.ac.vn; 5Faculty of Engineering Physics and Nanotechnology, VNU University of Engineering and Technology, 144 Xuan Thuy, Cau Giay, Hanoi 10000, Vietnam; anhht2508@gmail.com

**Keywords:** Fe_3_O_4_ poly-nanocrystals, silver nanopetals, Fe_3_O_4_-Ag architecture, magnetically assisted SERS sensing, rhodamine 6G

## Abstract

The formation of silver nanopetal-Fe_3_O_4_ poly-nanocrystals assemblies and the use of the resulting hetero-nanostructures as active substrates for Surface Enhanced Raman Spectroscopy (SERS) application are here reported. In practice, about 180 nm sized polyol-made Fe_3_O_4_ spheres, constituted by 10 nm sized crystals, were functionalized by (3-aminopropyl)triethoxysilane (APTES) to become positively charged, which can then electrostatically interact with negatively charged silver seeds. Silver petals were formed by seed-mediated growth in presence of Ag^+^ cations and self-assembly, using L-ascorbic acid (L-AA) and polyvinyl pyrrolidone (PVP) as mid-reducing and stabilizing agents, respectively. The resulting plasmonic structure provides a rough surface with plenty of hot spots able to locally enhance significantly any applied electrical field. Additionally, they exhibited a high enough saturation magnetization with M_s_ = 9.7 emu g^−1^ to be reversibly collected by an external magnetic field, which shortened the detection time. The plasmonic property makes the engineered Fe_3_O_4_-Ag architectures particularly valuable for magnetically assisted ultra-sensitive SERS sensing. This was unambiguously established through the successful detection, in water, of traces, (down to 10^−10^ M) of Rhodamine 6G (R6G), at room temperature.

## 1. Introduction

Surface-Enhanced Raman Spectroscopy (SERS) is one of the most popular optical-based analytical techniques for the detection of chemical and biological molecular traces [1,2,3,4,5]. SERS performed using sensitive substrates can provide specific fingerprint information for molecular recognition at the single molecule level. According to the SERS mechanism, rough surfaces including tips, vertex regions of plasmonic nanostructures, possessing more free electrons, can generate stronger localized surface plasmonic resonance (SPR) compared to smooth ones. The oscillation of free electrons in such regions are excited by an electromagnetic field such laser light, leading to a significant enhancement of the localized electromagnetic field for high SERS activity. Thus, many approaches have been developed to synthesize such anisotropic structures as nanostars, nanorods, nanotriangles, nanoplates, or nanosheets of gold [6,7,8], silver [9,10,11,12], or a mix of them [13,14]. For SERS applications, the integration of iron oxide component to these plasmonic structures is receiving great attention [15,16,17] since they offer the opportunity to selectively enrich the target molecules and to separate magnetically the desired substrate from the whole matrix. The detection procedure becomes simple and facilitates the cyclical use of the resulting SERS substrate.

Many approaches have been proposed focusing on the growth of an anisotropic plasmonic metal shell on an iron oxide core. One of them consists of a pre-coating of the iron oxide cores by a non-magnetic thin shell, like silica [18], graphite [19], graphene oxide [20], or titania [21] among others, before the attachment of noble metal seeds attachment and the further continuous noble metal shell growth. Despite this method is chemically effective, it leads to a net decrease of the whole magnetization due to the strong diamagnetic inter-shell contribution. This may be avoided by attaching directly, covalently, or electrostatically, preformed Ag or Au nanocrystals, which are then subsequently used to promote anisotropic metal seed-mediated growth. In this case, weak reducing agents such as L-ascorbic acid or hydroquinone were often required [22]. Polymer dispersing agents like polyvinyl pyrrolidone (PVP) introduced during the metal growth may also act as a structure-directing agent to control the final metal morphology [22]. The role of halides, in promoting a strongly crystal-facet dependent metal growth, was also highlighted, particularly when the reduction proceeds in presence of cationic surfactants such as cetyltrimethylammonium cation (CTA^+^) [23]. Halides may direct surfactant molecules adsorption and consequently metal nanocrystal anisotropic growth. Alternatively, an isotropic metal seed-mediated growth may be achieved and followed by a controlled etching step to reach the desired anisotropic metal morphology [24].

This non-exhaustive list of previous research papers is representative of the main existing material processing protocols in the relevant literature. So, in this study, we chose to focus on the Fe_3_O_4_-Ag system instead of the Fe_3_O_4_-Au, for which much more studies were already dedicated [22,25,26]. Some Fe_3_O_4_-Ag structures have been developed, consisting of core-satellites [27], Janus [28], microflowers [29], etc. These anisotropic structures were expected to present excellent SERS activities because rough surface and to be suitable for magnetic separation. However, the strategies for the fabrication of anisotropic silver on Fe_3_O_4_ surface are still a challenge, such types of structures have been uncommonly reported.

We prepared highly magnetized poly- and nano-crystalline magnetic oxide cores surrounded by an anisotropic in-shape plasmonic shell, rough enough to provide a great number of hot spots able to be used as smart substrates for magnetically assisted ultra-sensitive SERS detection. As a case study, we report here the SERS results we obtained on Rhodamine 6G (R6G) trace detection in water at room temperature (down to 10^−10^ M). In brief, our magneto-plasmonic architectures were provided by a facile synthesis of submicrometer-sized Fe_3_O_4_ particles in polyol, followed by grafting ammonium groups on their surface, through silane chemistry, to make them positively charged, and promoting thus electrostatic bonding to negatively charged ultra-small silver nanocrystals. The formation of the Ag petals in a 3D-dimensional plane onto the iron oxide cores was carried out through seed-mediated and self-assembly processes by reducing the dissolved AgNO_3_ salt with L-ascorbic acid in the presence of PVP. The achieved microstructural characteristics of these engineered Fe_3_O_4_-Ag nanocomposites were found to be suitable for the desired application: (a) They possess a strong magnetic responsiveness, allowing their magnetic collection by a lab magnet when contacted in a solution with a given analyte, allowing their use as SERS platforms; (b) They possess a multi-hot spots surface allowing a significant improvement of the Raman signature of the adsorbed analyte.

## 2. Materials and Methods

### 2.1. Materials

Silver nitrate (AgNO_3_, 99%), sodium borohydride (NaBH_4_, 98%), and rhodamine 6G (C_28_H_31_N_2_O_3_Cl, 99%) were purchased from Sigma-Aldrich. L-ascorbic acid (Vitamin C, 99.7%), polyvinyl pyrrolidone (PVP, M_w_~10,000 g mol^−1^), and absolute ethanol (98%) were purchased from Merck. Deionized water was used all preparations.

### 2.2. Synthesis of Fe_3_O_4_-Ag Seeds

Poly- and nanocrystalline Fe_3_O_4_ particles were prepared using the so-called polyol process (Figure 1) [30]. The resulting particles, of 180 nm in size, are composed of regular shaped tiny primary crystals of ~10 nm in size each (Appendix A). From zeta potentiometry measurements, we can conclude that their functionalization with APTES was successfully achieved using the protocol described by Nguyen et al. [30]. Indeed, zeta potential values of as prepared Fe_3_O_4_ particles and those functionalized with APTES were found to be +1.81 mV and +21.7 mV, respectively, in water (Appendix A).

Alongside the preparation of magnetic cores, ultra-small citrate coated Ag nanocrystals were prepared by reducing Ag^+^ by NaBH_4_ in water. In practice, 3 mL of trisodium citrate solution (1%) were added to 80 mL of silver nitrate solution (0.01 M). 0.1 mL of NaBH_4_ solution (2 M) was then dropped under stirring for 10 min. The silver seeds form yellow colloid, which was stored at 4 °C prior to use. The combination of citate-functionalized Ag nanoparticles and amine-functionalized Fe_3_O_4_ cores was based on a simple assembly process through electrostatic interactions (Figure 2). In the detail, 0.2 g of amine functionalized Fe_3_O_4_ particles were dispersed, by sonication, into 10 mL of deionized water and the resulting suspension was added into the formerly prepared silver colloid, and mechanically stirred for 1 h. The resulting Fe_3_O_4_-Ag seeds were then separated by magnetic decantation, washed several times with deionized water and dispersed in deionized water for a subsequent use.

The successful attachment of the negatively charged Ag nanoparticles to the positively charged Fe_3_O_4_ ones was corroborated by UV spectroscopy: one could notice the appearance of a shoulder at 400 nm, characteristic of the Ag SPR band in the free Ag spectrum (Figure 2). The silver nanocrystals attachment was underlined by the change in the surface charge of +21.7 mV and −17.7 mV for Fe_3_O_4_ coated with APTES and Fe_3_O_4_-Ag seeds, respectively (Appendix A).

### 2.3. Synthesis of Fe_3_O_4_-Ag Nanocomposites

Fe_3_O_4_-Ag nanocomposites were prepared by the reduction of a fresh AgNO_3_ aqueous solution in the presence of as-prepared Fe_3_O_4_-Ag seeds, PVP, and L-ascorbic acid. The concentrations of PVP, AgNO_3_, and L-ascorbic acid remain unchanged, while the amount of Fe_3_O_4_-Ag seeds was varied. AgNO_3_/Fe_3_O_4_-Ag seeds weight ratio, quoted *r*, was varied from 10 to 2, maintaining the reaction time constant whatever the value of *r*.

Then, 1 to 5 mg of Fe_3_O_4_-Ag seeds and 0.5 mg of PVP were added to 2 mL of deionized water. The mixture was sonicated and stirred to ensure the dispersion of the particles, and 3.5 mg of L-ascorbic acid was then dissolved into the mixture. Finally, 0.6 mL of an aqueous solution of AgNO_3_ (0.1 M) was added. The suspension was mechanically stirred continuously for 10 min to ensure the completion of the reaction. The resulting nanocomposites were recovered by magnetic decantation, washed several times with deionized water and dried in air. The same experiments were repeated without adding PVP to the reaction medium and the recovered powders were used as references.

### 2.4. Preparation of SERS Substrates

About 50 µL (0.5 mg) of as-prepared nanocomposites, prepared with or without PVP, were dispersed in 2 mL of analyte solution for 5 h to ensure the adsorption of the analyte molecules on their surface. The analyte solution consisted of R6G diluted in deionized water at different concentrations equal to 10^−5^, 10^−6^, 10^−7^, 10^−8^, 10^−9^, and 10^−10^ M. Finally, the magnetic nanocomposites were collected by a magnet and transferred to a clean glass surface; they were washed with deionized water while maintained by the magnet on the glass, then dried at 60 °C for 10 min prior to their analysis by SERS.

### 2.5. Characterizations

SEM was performed on a S-4800 microscope (Hitachi, Kyoto, Japan) applying a voltage of 5 kV. A drop of the sample solution was deposited on the silicon wafer and drying at room temperature. XRD was carried out using a EQUINOX 5000 (Thermo Scientific, Illkirch-Graffenstaden, France) in a θ–θ Bragg–Brentano reflection configuration in the 20–80° 2θ angular range. UV–Vis absorption spectroscopy was achieved using a UV2600 spectrophotometer (Shimadzu, Kyoto, Japan). Absorbance spectra were recorded for wavelengths from 300 nm to 800 nm. DC magnetometry was performed at room temperature on a 7404-VSM vibrating sample magnetometer (Lake Shore, Westerville, OH, USA). In practice, a given mass of each sample is compacted in a diamagnetic sampling tube and introduced into the magnetometer. SERS measurements were conducted on a Raman spectrometer using a 30 mW laser source with an excitation wavelength of 633 nm and light spot diameter of 10 µm. All spectra were taken with an exposure time of 1 s and 10 accumulations, and recorded within the 300–2000 cm^−1^ spectral range. For each sample, an average of three SERS spectra of the substrate was carried out.

## 3. Results

### 3.1. Phase and Microstructure Analysis of the Engineered Fe_3_O_4_-Ag Nanocomposites

The crystal structures of the produced composite particles were checked by XRD. Whatever the *r* value, using or not PVP during the synthesis, the recorded patterns exhibit two sets of diffraction peaks: a first set with highly intense peaks at 2ϴ positions of 38.31, 44.41, 64.32, and 77.12°, corresponding to the (111), (200), (220), (311), and (222) planes of the cubic face-centered Ag phase, and a second set of peaks with a very weak intensity (Figure 3), particularly at 2ϴ positions of 35.23°, attributed to the (331) planes of the spinel Fe_3_O_4_ phase. This peak intensity difference is attributed to the large electronic density of the metallic phase compared to the oxide one. Note that varying the *r* value does not significantly change the appearance of the patterns: only the intensity of the spinel diffraction line decreases when *r* increases.

The morphology of the nanocomposites was imaged by SEM. Representative micrographs are given hereafter to illustrate the general shape of the silver shell (Figure 4). Interestingly, the particles produced without PVP appear to be almost isotropic. Rather spherical sub-micrometer-sized particles exhibiting a smooth surface seem to constitute the powders obtained in these conditions: the growth of a continuous Ag shell was achieved by combining molecular self–assembly technique and seed-mediated growth, which took place around Ag seeds present on the surface of Fe_3_O_4_ cores. By increasing the *r* value, a continuous shell progressively covers the surface of all magnetic cores (Figure 4a,b). Reversely, in presence of PVP, the morphology completely changed. A more and more irregular silver structure grows around the spherical magnetic iron oxide cores, as the *r* value increases, leading to the development of 3D-petals (Figure 4c,d). The size of Ag petals increases from around 300 to 500 nm when the *r* value increases from 2 to 10, without any change in their petal-like shape. These significant morphological differences between the two-sample series may be explained by the role played by PVP during the fast reduction of a great amount of AgNO_3_ by L-ascorbic acid, leading to seed-mediated growth of silver on Fe_3_O_4_-Ag seed surface rather than silver-nucleation. We suggest that PVP, preferentially adsorbed to the high index silver primary particle faces, mainly the {111} ones, drove oriented self-assembly of these particles, leading to percolating Ag petals on Fe_3_O_4_-Ag seed surface. Without PVP, this aggregation was completely random, leading to an isotropic silver growth. EDX was performed on the samples to quantify their Fe and Ag contents. All the recorded spectra evidenced the characteristic peaks of not only Fe and Ag elements, but also of Si from APTES and silicon grid, and C from PVP residue. As expected, the Fe content increased when the ratio *r* decreased (Appendix A and Appendix A).

UV–Visible absorption spectra of the aqueous suspensions of Fe_3_O_4_-Ag nanocomposites confirmed this microstructural difference between the samples prepared with and without PVP. Indeed, all the spectra exhibit a strong and broad absorption related to the plasmonic properties of the silver component (Figure 5); however, in the presence of PVP, absorption was observed for a wavelength of 433 nm while it was observed at 455 nm without using PVP. Moreover, an additional sharp peak appeared at about 343 nm for this last sample series, in good agreement with a plate silver structure [31].

The magnetic properties of all the produced Fe_3_O_4_ particles and their related nanocomposites were investigated using VSM magnetometer. The variation of their magnetization as a function of the magnetic field M(H) was measured at room temperature (RT) (Figure 6). The two hysteresis loops were found to be similar, presenting a non-zero but low remanence (<2 emu g^−1^) and coercivity (<21 Oe) typical of soft ferrimagnets. The magnetization appeared to follow quite reversibly the magnetic field, meaning that they can be easily attracted by applying an external, and even small in magnitude, dc-magnetic field. The saturation magnetization of Fe_3_O_4_-Ag seeds was found to be 70.9 emu g^−1^, a value close to that reported for bare Fe_3_O_4_, and it decreased down to 8.9, 36.8, and 9.7 emu g^−1^ in the Fe_3_O_4_-Ag nanocomposites produced without PVP for a *r* value of 10, with PVP for the different *r* values of 2 and 10, respectively, due to the diamagnetic silver shell contribution.

Obviously, the higher the amount of silver contained in the nanocomposites, the lower the saturation magnetization; however, it remains satisfactory with regard to the use of samples in magnetic assisted SERS application.

### 3.2. Application in SERS

Fe_3_O_4_-Ag nanocomposites produced using PVP and presenting Ag petal-like shell with a rough surface, are expected to induce a high SERS effect. First, the assisted SERS detection assays were performed with all nanocomposites to determine the best SERS response. In practice, analyte aqueous solutions containing a given contaminant, R6G, at a concentration of 10^−6^ M, were prepared, in which Fe_3_O_4_, Fe_3_O_4_-seeds, and Fe_3_O_4_-Ag particles were dispersed to get stable suspensions (at least during the contacting to analyte time). They were then collected onto SERS substrates by applying an external magnetic field. Homogeneously distributed aggregates were formed spontaneously on the substrate wafer after water washing and drying. The SERS measurements were subsequently carried out on the as-prepared SERS substrates (Appendix A). Note that Raman spectra were carried out by the average of several collections, for each starting analyte solution, on different positions randomly selected on the surface of Fe_3_O_4_-Ag based SERS substrates. Characteristics R6G peaks positioned at 612, 774, and 1185 cm^−1^ corresponding to the C-C-C ring in-plane bending, C-H out-of-plane bending, C-H in-plane bending, and at 1313, 1363, 1510, and 1654 cm^−1^ corresponding to aromatic C-C stretching, are clearly evidenced in the collected spectra. Raman spectra performed on Fe_3_O_4_-Ag nanocomposites, produced using PVP for a *r* value of 10, presented the best SERS signal compared to the particles produced using PVP for a *r* value of 2, or without PVP. Indeed, the highest the *r* value, the highest the SERS activity, due the more anisotropic morphology that was developed using a higher amount of Ag in presence of PVP.

The concentration of R6G was changed to test Fe_3_O_4_-Ag nanocomposites produced with PVP for values of 10 (Figure 7a). Characteristics R6G peaks were also clearly evidenced in the collected Raman spectra; whatever the R6G concentration in the starting analyte solution, traces down to 10^−10^ M of R6G could be detected. The reproducibility of the SERS substrate is presented in Figure 7b. SERS signals were investigated on a large area, collected randomly 11 times on the substrate. The relative standard deviations (RSD) of SERS intensities at 1510 cm^−1^ was found to be 16% (Appendix A), which led us to conclude to the uniformity and reproducibility of these measurements.

Focusing on the most intense R6G peak, centered at 1510 cm^−1^, the plot of the log of its intensity as a function of the log of R6G concentration (Figure 8) shows a quantification region down to 10^−10^ M with a linear relationship, which can be expressed by Equation (1) and an analytical enhancement factor (AEF) can be deduced using Equation (2):log(I_1510_) = 0.298 × log[R6G] + 6.238(1)
AEF = (I_SERS_/C_SERS_)/(I_RS_/C_RS_)(2)
where I_SERS_ is the SERS signal at C_SERS_ concentration of R6G and I_RS_ is the Raman signal under non-SERS conditions at C_RS_ concentration of R6G. By applying Equations (1) and (2), an AEF of 3 × 10^7^ was found for the engineered magnetic SERS substrates developed in this study. The resulting value is a fairly good compared to SERS substrates fabricated by spherical silver nanoparticles [32,33]. However, the whole SERS community assumed that it is a good qualitative indication of the opportunity to routinely use such a system for rapid trace detection [34]. As a consequence, the engineered Fe_3_O_4_-Ag nanocomposites we developed there are clearly good candidates for a sensitive and rapid organic pollutant (e.g., pesticides, and antibiotics) SERS detection.

## 4. Conclusions

We developed a strategy to prepare Fe_3_O_4_-Ag nanocomposites for SERS application. Polyol-made submicrometer-sized poly and nanocrystalline Fe_3_O_4_ particles were synthesized through a simple and one-step method. A subsequent Ag petal-like shell growth onto Fe_3_O_4_ platforms was carried out through seed-mediated and self-assembly growth by using L-ascorbic acid as mid-reducing agent in a presence of PVP. The resulting hetero-nanostructures exhibited a high enough saturation magnetization (M_s_ = 9.7 emu g^−1^), allowing their reversible collection by an external magnetic field, even small in magnitude, at room temperature, as well as a quite rough plasmonic surface with a dense and homogeneous hot spot distribution to serve as magnetically assisted SERS sensors. Indeed, contacted with a R6G analyte solution of different concentrations and then separated by a magnet and deposited on a glass wafer, on which R6G Raman spectra were collected, they evidenced a net R6G SERS detection capability even when starting from a very diluted analyte solution (down to 10^−10^ M).

## Figures and Tables

**Figure 1 nanomaterials-11-01288-f001:**
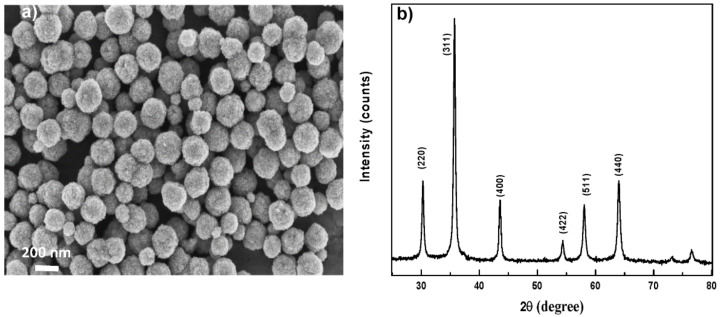
(**a**) Scanning electron microscopy (SEM) image of an assembly of Fe_3_O_4_ particles and (**b**) X-ray diffraction (XRD) pattern. It must be noticed that Fe_3_O_4_ particles exhibit an average size of 180 ± 20 nm and they are composed of regular shaped tiny primary crystals of 10 nm in size. The XRD pattern matches very well with the spinel structure, with a Rietveld refined cell parameter of 8.390 ± 0.005 Å, very close to that of pure magnetite (ICDD n°98-002-6410).

**Figure 2 nanomaterials-11-01288-f002:**
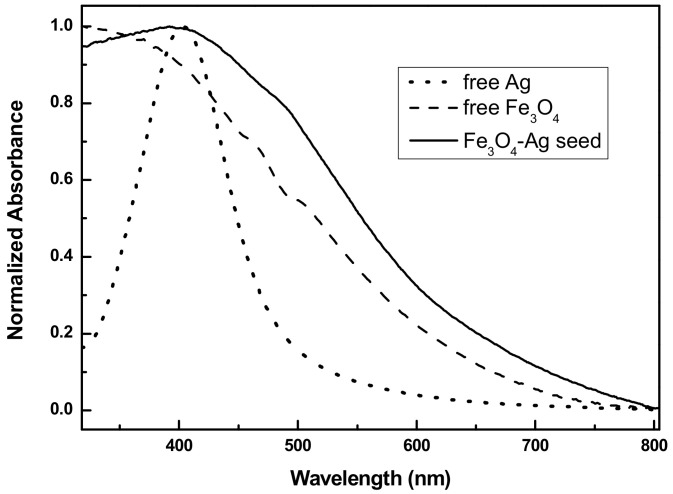
Normalized UV–Visible absorption spectra, in a transmission mode, of as-prepared Ag seeds (dotted line), Fe_3_O_4_ cores (dashed line) and Fe_3_O_4_-Ag seeds (continuous line) aqueous suspensions.

**Figure 3 nanomaterials-11-01288-f003:**
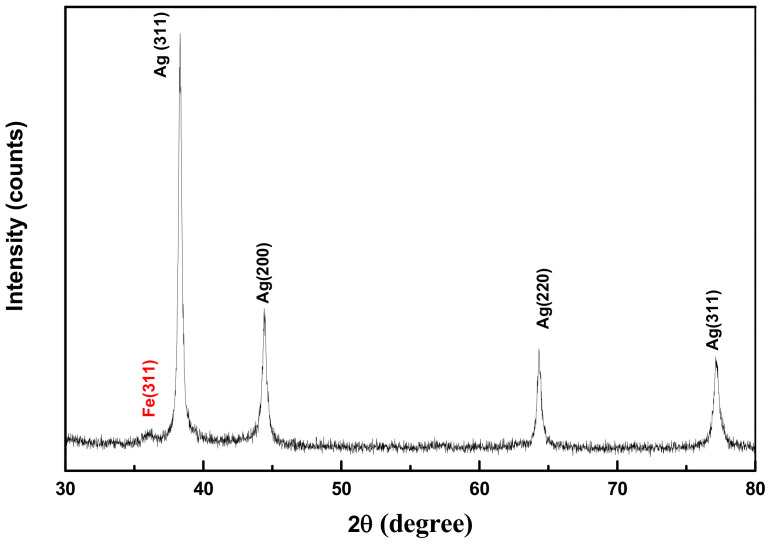
XRD pattern of a representative Fe_3_O_4_-Ag nanocomposite sample (*r* = 10), indexed as a mix of the metallic silver and the iron oxide phases.

**Figure 4 nanomaterials-11-01288-f004:**
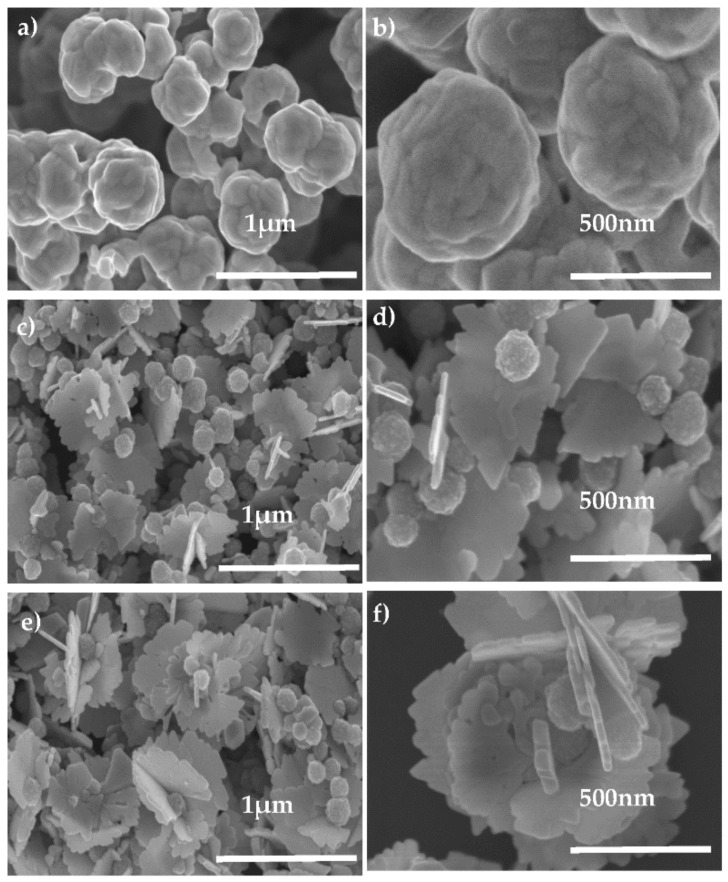
SEM images of Fe_3_O_4_-Ag nanocomposites produced (**a**) without PVP for a *r* value of 10, (**c**,**e**) with PVP for a *r* value of 2 and 10, respectively. To highlight the isotropic and anisotropic silver shell morphology, a zoom is given in (**b**,**d**,**f**), respectively.

**Figure 5 nanomaterials-11-01288-f005:**
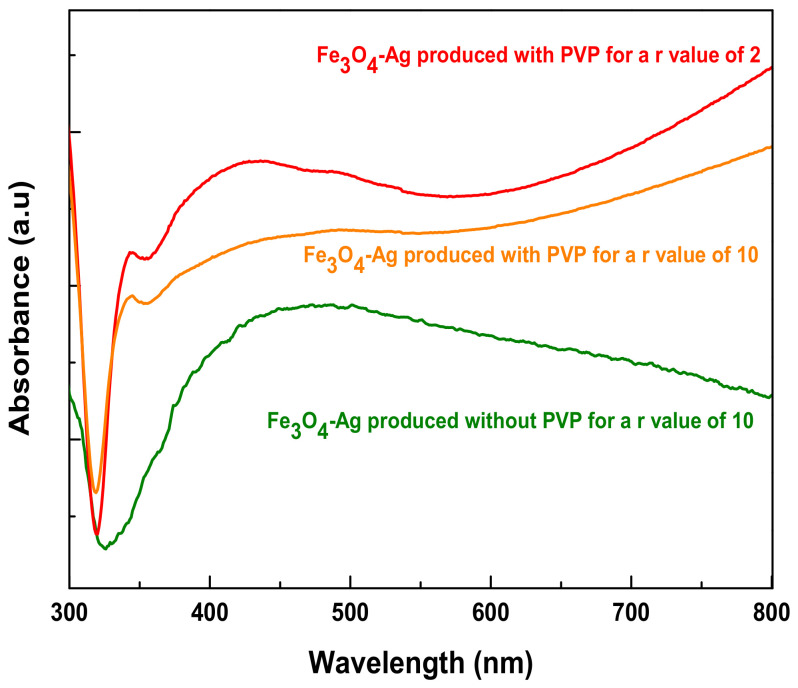
UV–Vis spectra of Fe_3_O_4_-Ag nanocomposites produced without PVP with a *r* value of 10 (green line) compared to those prepared in presence of PVP for different *r* values, *r* = 2 (red line) and *r* = 10 (orange line).

**Figure 6 nanomaterials-11-01288-f006:**
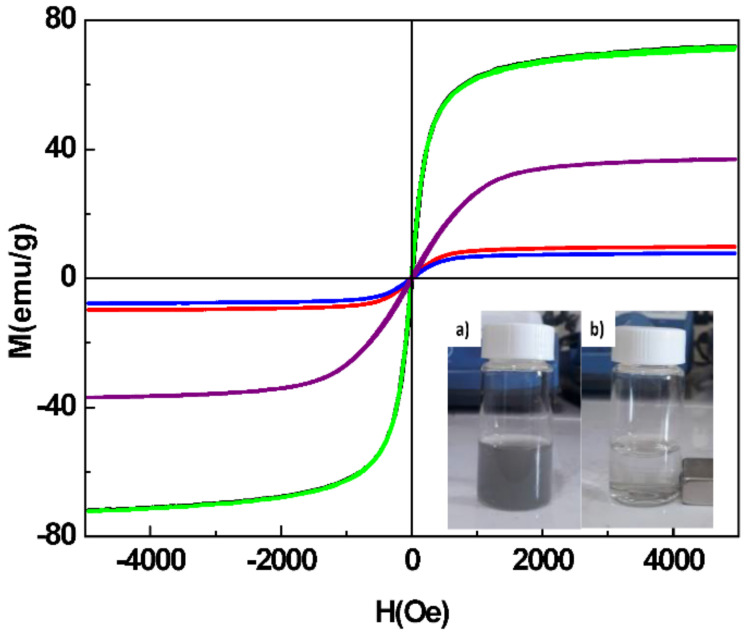
Magnetization curves measured at 300 K on bare Fe_3_O_4_ particles (black line), Fe_3_O_4_-Ag seeds (green line) and their Fe_3_O_4_-Ag related nanocomposites produced in presence of PVP for a different *r* value of 10 (red line) and 2 (purple line); without PVP (*r* = 10) (blue line). In the inset, a photograph of Fe_3_O_4_-Ag nanocomposite (with PVP, *r* = 10) suspension in deionized water (a) before and (b) after magnet separation.

**Figure 7 nanomaterials-11-01288-f007:**
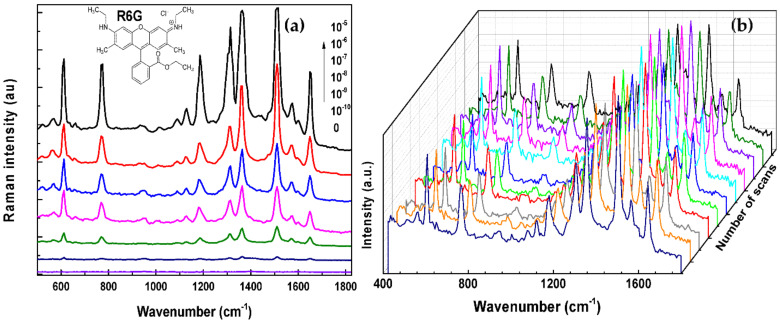
(**a**) SERS spectra collected on the engineered substrates after exposition to R6G aqueous solutions of different concentrations. (**b**) Series of SERS spectra collected on a single substrate after exposition to a R6G 10^−9^ M aqueous solution. The measurements were performed on different areas on the wafer. All spectra were taken with an exposure time of 1 s and 10 accumulations.

**Figure 8 nanomaterials-11-01288-f008:**
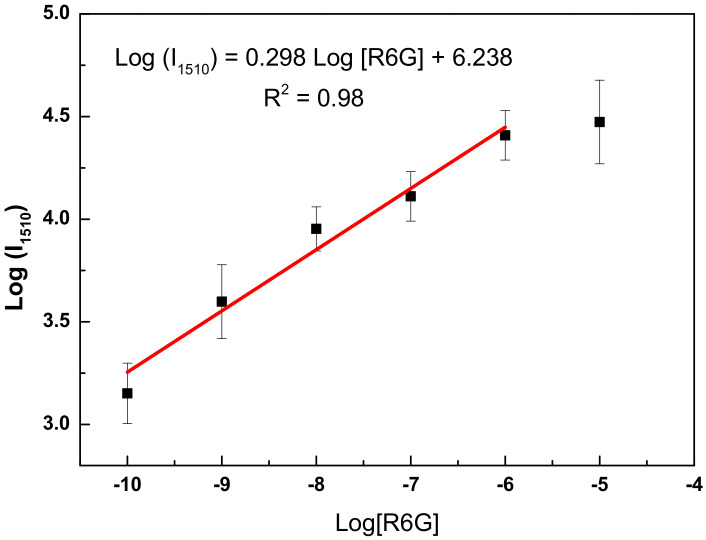
The functional relationship between log of Raman intensity at 1510 cm^−1^ and log of R6G concentration. The error bars are standard deviations from a total of three measurements.

## Data Availability

The data presented in this article are available on request from the corresponding authors.

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
