# Peer review of "Preparation of Fe3O4-Ag Nanocomposites with Silver Petals for SERS Application"

_nanomaterials, 2021, doi:10.3390/nano11051288_

Round 1

Reviewer 1 Report

Abstract section should be revised by inserting some key results, while keeping only the relevant information.

Figure 5. Please replace Absorption with Absorbance.

Results and discussion: To increase the scientific value of the manuscript Authors should consider extension of the obtained results section with comparison of obtained results with the results described in previous publications.

Also, the language of the manuscript has to be considerably/ technically improved. The current version of text contains a number of language issues.

Finally, I consider that the manuscript requires a major revision before resubmitting.

Author Response

Dear Reviewer,

We want to thank you for reviewing our manuscripts. We appreciate constructive comments. We have addressed the feedback and improved our manuscript. All our response of your comments was described with red colour as following:

(1) Abstract section should be revised by inserting some key results, while keeping only the relevant information.

As suggested by referee, the abstract section now added some key results.

(2) Figure 5. Please replace Absorption with Absorbance.

It has been corrected

(3) Results and discussion: To increase the scientific value of the manuscript Authors should consider extension of the obtained results section with comparison of obtained results with the results described in previous publications.

As suggested by the referee, some results were now added to increase the scientific value in Supporting information. The references [30-34] were also added in the results to compare obtained results with literature reported values

(4) Also, the language of the manuscript has to be considerably/ technically improved. The current version of text contains a number of language issues.

The language of manuscripts was improved

Sincerely,

Reviewer 2 Report

Nanomaterials-1177311

The manuscript entitles “Preparation of Fe3O4-Ag nanocomposites with silver petals for SERS application” by Nguyen et al. lacks in terms of material characterization and SERS analysis.

After a careful look in the literature, there are several papers demonstrating the magnetic separation as a technique to enrich the analyte in the SERS platform to get a better signal. The authors must highlight the new/novelty/innovation of these materials in comparison to the ones reported in the literature.

I do not recommend publication after major revisions.

See comments below:

  1. Why reference 6, 7 and 8 have lots of referenced papers and the others only have one? Please use the same methodology in all references.
  2. Can the authors explain why they denominate the Fe3O4 particles as poly- and nanocrystalline? They seem very monodispersed to me.
  3. Can the authors present a SEM or TEM image with higher resolution of Figure 1 for the readers see the tiny primary crystals of 10 nm?
  4. The functionalization of the Fe3O4 particles, which is a crucial step for the fabrication of the Ag petals, should be demonstrated using FTIR, elemental analysis and zeta potential.
  5. What is the amount of Silver in each composite (including the Fe3O4-Ag seeds)?
  6. What is the amount of composite used in the SERS analysis? It was in solid or solution? This must be in the experimental section
  7. Why the authors assumed that the most marked silver petal-like morphology will have the better SERS behavior, before performing the SERS analysis (Page 7, line 210)? Please comment
  8. The variation of the magnetization as a function of the magnetic field should be performed to all composites to understand the contribution of the Ag NPs.
  9. The authors claim in the Applications in SERS section “Based on our previous results and as already said previously,…”. What the authors mean with previous results?
  10. For the SERS analysis, the authors should perform comparative studies:
  11. Compare the SERS signal of R6G using the composite r=10 with and without magnetic separation of the analyte
  12. Compare the SERS signal of R6G using all the composites (with and without PVP and the Fe3O4-Ag seeds)
  13. Compare the SERS signal of R6G using the composite with r=10, only the Fe3O4 particles and Ag nanoparticles prepared by the same method without the Fe3O4.
  14. Conventional Raman spectra of the composites should also be demonstrated.
  15. How many replicas and points were analyzed to perform Figure 8? This should be described in the experimental part
  16. Figure 7b, I believe that 6 spectra are a very little sample to indicate the reproducibility of the SERS signal using the composite with r=10. This should be performed with more points. What is the RDS value for the band 1510 cm-1?
  17. The bands for R6G should be assigned in the manuscript.
  18. The authors claim that these materials are good candidates for sensitive and rapid organic pollutants, like pesticides or antibiotics, however, they only demonstrate the SERS sensitivity for an organic dye which as a very good SERS signal. Please comment.

Author Response

Dear Reviewer,

We want to thank you for reviewing our manuscripts. We appreciate constructive comments. We have addressed the feedback and improved our manuscript. All our response of your comments was described with red colour as following:

  1. Why reference 6, 7 and 8 have lots of referenced papers and the others only have one? Please use the same methodology in all references.

It has been changed. We are sorry for this inconvenience.

  1. Can the authors explain why they denominate the Fe3O4particles as poly- and nanocrystalline? They seem very monodispersed to me.

Fe3O4 particles are composed of regular shaped tiny primary crystals with a size of ~10 nm, while their total final size of180 nm

  1. Can the authors present a SEM or TEM image with higher resolution of Figure 1 for the readers see the tiny primary crystals of 10 nm?

SEM and TEM of Fe3O4 were added in supporting information (Figure S1). So, we can see regular shaped tiny primary crystals with a size of ~10 nm

  1. The functionalization of the Fe3Oparticles, which is a crucial step for the fabrication of the Ag petals, should be demonstrated using FTIR, elemental analysis and zeta potential.

The functionalization of the Fe3Oparticles was demonstrated by XPS and zeta potential in our previous papers (Nguyen, T. T.; Lau-Truong, S.; Mammeri, F.; Ammar, S., Star-Shaped Fe3-xO4-Au Core-Shell Nanoparticles: From Synthesis to SERS Application. Nanomaterials 2020, 10(2), 294).

  1. What is the amount of Silver in each composite (including the Fe3O4-Ag seeds)?

The amount of silver in each composite were determined by EDS in supporting information (Figure S2)

  1. What is the amount of composite used in the SERS analysis? It was in solid or solution? This must be in the experimental section

The amount of composite used in SERS analysis was about 0.5 mg in solution. It was added in the experimental section

  1. Why the authors assumed that the most marked silver petal-like morphology will have the better SERS behavior, before performing the SERS analysis (Page 7, line 210)? Please comment

The silver petal-like morphology exhibit a broad range of wavelengths. Raman signals could be highly localized in spatially narrow regions such as sharp corners and edges of the petal-like. Some recent experimental papers proved that Ag nanopetal (nanosheets) could be utilized as desired substrates for SERS application. Liu, Guangqiang, et al. "Standing Ag nanoplate-built hollow microsphere arrays: Controllable structural parameters and strong SERS performances." Journal of Materials Chemistry 22.7 (2012): 3177-3184.

  1. The variation of the magnetization as a function of the magnetic field should be performed to all composites to understand the contribution of the Ag NPs.

To determine the contribution of the Ag components, EDS results were added in supporting information (Figure S2).

  1. The authors claim in the Applications in SERS section “Based on our previous results and as already said previously,…”. What the authors mean with previous results?

The text has been removed

  1. For the SERS analysis, the authors should perform comparative studies:
  2. Compare the SERS signal of R6G using the composite r=10 with and without magnetic separation of the analyte

There are no change between SERS signal of R6G using the composite with magnetic separation and without magnetic separation of the analyte. Several papers demonstrated the magnetic separation as a technique to enrich the analyte in the SERS platform to get a better signal  

  1. Compare the SERS signal of R6G using all the composites (with and without PVP and the Fe3O4-Ag seeds)

The compasion the SERS signal of R6G using all the composites was added in Figure S4 (Supporting information)

  1. Compare the SERS signal of R6G using the composite with r=10, only the Fe3O4particles and Ag nanoparticles prepared by the same method without the Fe3O4.

The compasion the SERS signal of R6G using all the composites was added in Figure S4 (Supporting information). Ag particles synthezied by the same method will be have different morphology compare to using Fe3O4. SERS signal can be better but magnetic properties can be used for separation and pre-concentration of the analytes  

  1. Conventional Raman spectra of the composites should also be demonstrated.

Conventional Raman spectra of the composites were shown in Figure 7a (with R6G 0M)

  1. How many replicas and points were analyzed to perform Figure 8? This should be described in the experimental part

There are three poins which analyzed in Figure 8. It was added in the experimental part

  1. Figure 7b, I believe that 6 spectra are a very little sample to indicate the reproducibility of the SERS signal using the composite with r=10. This should be performed with more points. What is the RDS value for the band 1510 cm-1?

Some more points were added in Figure 7b. The residual sum of squares value is

  1. The bands for R6G should be assigned in the manuscript.

The bands for R6G were added in Table S2 (Supporting information)

  1. The authors claim that these materials are good candidates for sensitive and rapid organic pollutants, like pesticides or antibiotics, however, they only demonstrate the SERS sensitivity for an organic dye which as a very good SERS signal. Please comment.

The paper demonstred the SERS sensitivity for R6G dye. Some organic pollutants also can be detected by using Fe3O4-Ag Nanocomposites.

Sincerely,

Reviewer 3 Report

This paper presents the possibility of Iron oxide-Silver composites with silver petals for SERS application. Their experimental results are showing good sensitivity of SERS sensors for Rhodamine 6G. So I recommend this paper to be published in the journal.

Author Response

Dear Reviewer,

We want to thank you for reviewing our manuscript.

Sincerely,

Reviewer 4 Report

In this paper “Preparation of Fe3O4-Ag Nanocomposites With Silver Petals for SERS Application” Authors presents methodology formation of silver nanopetal-Fe3O4 poly-nanocrystals assemblies and their use of the resulting hetero-nanostructures as active substrates for Surface Enhanced Raman Spectroscopy (SERS) application. Produced plasmonic structure provides a rough surface with plenty of hot spots able to locally enhance significantly any applied electrical field.

In my opinion this paper can be interesting to readers of Nanomaterials journal. The paper is clearly presented. The paper contains 8 figures – figures are legible and good quality.

English of the paper is rather good – in my opinion the language of the paper should be a little improved. I am asking for corrections by a native speaker.

I find some mistakes for example:

  • Introduction chapter – Authors should include new information about Fe3O4-Ag Nanocomposites from modern (from last years) papers of global research.
  • Materials and Methods chapter – Please describe equipment used in the experiment – work development environment / work apparatus should be given – model of equipment (manufacturer, city, country).
  • Results chapter – should include a discussion related to the topic of the paper (expansion of research methodology, presentation of research results obtained).
  • In my opinion Conclusions chapter should be a little changed. In this chapter there are no summary of all significant information obtained by the Authors and written in the Results
  • In-text citations - please use citations as described in the article preparation guidelines - use square brackets "[...]"
  • Please prepare a literature review according to the guidelines of the Nanomaterials journal.
  • Amount of references is also sufficient but some papers cited in the references 29 from all 43 are older then 5 years – these publications constitute over 67 % of all cited papers. I propose to add some new (from the last 5 years) publications on the production or properties of these type of materials. Authors should include several modern papers of global research in this field.
  • Minimum 30 papers from all 43 (over 69 %) are wrote by authors from Asia – China, Vietnam, India and others. Are the topics covered in this article only by research centers from Asia? I ask for a reliable verification of global research in this field. I propose to add some new (from the last 5 years) publications on the production or properties of these type of materials. Authors should include several modern papers (also from Europe and America).
  • In the list of references I found only 2 papers of the Authors of reviewed paper. Are these the first studies related to this subject for the Authors?
  • In Figure 7b – no details are visible – please increase the size of descriptions on the axes.
  • When the Authors mean – "… research works ..." (for example line 69) the phrase "… research papers ..." or "… research articles ..." should be used, but not the word "… work …".
  • In the whole paper, you write the values in percent as for example 99% (for example line 91) – you should write and value with unit with spaces (99 %).

The results obtained are interesting and promising. The manuscript can be accepted for publication in Nanomaterials journal after MAJOR corrections.

Author Response

Dear Reviewer,

We want to thank you for reviewing our manuscripts. We appreciate constructive comments. We have addressed the feedback and improved our manuscript. All our response of your comments was described with red colour as following:

  • Introduction chapter – Authors should include new information about Fe3O4-Ag Nanocomposites from modern (from last years) papers of global research.

Introduction chapter was added some new information about Fe3O4-Ag Nanocomposites in recent year

  • Materials and Methods chapter – Please describe equipment used in the experiment – work development environment / work apparatus should be given – model of equipment (manufacturer, city, country).

The description was added in the manuscripts

  • Results chapter – should include a discussion related to the topic of the paper (expansion of research methodology, presentation of research results obtained).

Some research methodologies such as EDS, Zeta potential were added in Supporting information to demonstrate the obtained results

  • In my opinion Conclusionschapter should be a little changed. In this chapter there are no summary of all significant information obtained by the Authors and written in the Results

Conclusions chapter was added a summary of the obtained results

  • In-text citations - please use citations as described in the article preparation guidelines - use square brackets "[...]"

Citation format was changed

  • Please prepare a literature review according to the guidelines of the Nanomaterials journal.

A literature review was prepared according to the guidelines of the Nanomaterials journal

  • Amount of references is also sufficient but some papers cited in the references 29 from all 43 are older then 5 years – these publications constitute over 67 % of all cited papers. I propose to add some new (from the last 5 years) publications on the production or properties of these type of materials. Authors should include several modern papers of global research in this field.

As suggested by the referee, several modern papers was added to our references

  • Minimum 30 papers from all 43 (over 69 %) are wrote by authors from Asia – China, Vietnam, India and others. Are the topics covered in this article only by research centers from Asia? I ask for a reliable verification of global research in this field. I propose to add some new (from the last 5 years) publications on the production or properties of these type of materials. Authors should include several modern papers (also from Europe and America).

As suggested by the referee, several modern papers was added to our references

  • In the list of references I found only 2 papers of the Authors of reviewed paper. Are these the first studies related to this subject for the Authors?

Yes, are there

  • In Figure 7b – no details are visible – please increase the size of descriptions on the axes.

Figure 7b was changed

  • When the Authors mean – "… research works ..." (for example line 69) the phrase "… research papers ..." or "… research articles..." should be used, but not the word "… work …".

The text has been corrected

  • In the whole paper, you write the values in percent as for example 99% (for example line 91) – you should write and value with unit with spaces (99 %).

The text has been corrected

Sincerely,

Round 2

Reviewer 1 Report

It is unacceptable to pretend that corrections were made when they were not. The following comments were discarded:

  1. Abstract section should be revised by inserting some key results, while keeping only the relevant information. Please highlight the performed changes.
  2. Results and discussion: To increase the scientific value of the manuscript, Authors should consider extension of the obtained results section with comparison of obtained results with the results described in previous publications. – The text is the same, but the references are different. This is confusing. Please explain. Furthermore, expression such as „This is a fairly good value regarding the existing literature...” are also unclear. Please provide more information.
  3. Also, the language of the manuscript has to be considerably/ technically improved. The current version of text contains a number of language issues.

Please refer to them, while highlighting the changes.

Also, line 74 Please define “…”.

Finally, I consider that the manuscript requires a major revision before resubmitting.

Author Response

Dear Reviewer,

We want to thank you for reviewing our manuscript.

Please see the attachment,

Sincerely,

Reviewer 2 Report

The authors did not answer all my questions, or the answer is vague.

There are still open questions, so I do not recommend publication after major revisions

  1. Why the figures and tables of the supporting information are not referenced in the manuscript? What is the point to put some data in the supporting information if the readers do not know the existence of that information? This should be revised. Comments on these data should be add to the manuscript.
  2. The zeta potential of the Fe3O4 should also be shown in Figure S3
  3. Question 8-The variation of the magnetization as a function of the magnetic field should be performed to all composites to understand the contribution of the Ag NPs. The authors reply - To determine the contribution of the Ag components, EDS results were added in supporting information (Figure S2).

What the authors mean with this answer? Even more, the Figure S2 is not referenced in the manuscript. I will try again; I want to know the magnetic behavior of the composites with different amounts of Ag NPs. Please present the magnetization curves for all composites (Fe3O4-Ag with PVP (r=2), Fe3O4-Ag without PVP (r=10) and Fe3O4 – Ag seeds).

  1. Question 11 – If there is no difference in the SERS signal of R6G using the composite with magnetic separation and without magnetic separation of the analyte, what is the point to perform the magnetic separation. The authors are saying that the effect of pre-concentration of the analyte using the magnetic separation does not occur, so why use Fe3O4 Or why use magnetic separation in the SERS studies if the result is the same? Please comment
  2. Question 13 – The authors answer: Ag particles synthezied by the same method will be have different morphology compare to using Fe3O4. SERS signal can be better but magnetic properties can be used for separation and pre-concentration of the analytes. However, the authors said that the SERS signal is the same with or without magnetic separation. Again, what is the point of using magnetite in this particular case? This should be very clear and highlighted in the manuscript.
  3. What is the relative standard deviation (RSD) for Figure 7b
  4. Figure S4: what is the different between Fe3O4-Ag nanocomposites with PVP (r = 10) align a and b?

Author Response

(The authors gave the same response as above.)

Reviewer 4 Report

In corrected paper "Preparation of Fe3O4-Ag Nanocomposites With Silver Petals for SERS Application" Authors have properly addressed the concerns from the referee. All my remarks have been included in the revised document. Below you will find my comments on the attached answers.

Authors reformatted and extended Introduction, Materials and Methods, Results and Conclusion chapter. Authors significantly reformat the entire article. Most of the unnecessary figures have been removed – the figures (suggested) has been changed in accordance with the recommendations. They modified a list of a references, too.

Referring to my substantive reservations – the authors made the necessary modifications. Authors have improved the language – language corrections are sufficient. Now, paper is preparing in accordance with the journal's guidelines.

The manuscript can be accepted for publication in Nanomaterials journal in the current form.

Author Response

(The authors gave the same response as above.)

Round 3

Reviewer 1 Report

The manuscript can be accepted in the current form.

Reviewer 2 Report

Accept in present form